# VTON-VLLM: Aligning Virtual Try-On Models with Human Preferences

**Siqi Wan**[1]*, **Jingwen Chen**[2], **Qi Cai**[2], **Yingwei Pan**[2], **Ting Yao**[2], **Tao Mei**[2]

University of Science and Technology of China[1],    [2] HiDream.ai

wansiqi4789@mail.ustc.edu.cn, {chenjingwen, cqcaiqi, pandy, tiyao, tmei}@hidream.ai

## Abstract

Diffusion models have yielded remarkable success in virtual try-on (VTON) task, yet they often fall short of fully meeting user expectations regarding visual quality and detail preservation. To alleviate this issue, we curate a dataset of synthesized VTON images annotated with human judgments across multiple perceptual criteria. A vision large language model (VLLM), namely VTON-VLLM, is then learnt on these annotations. VTON-VLLM functions as a unified "fashion expert" and is capable of both evaluating and steering VTON synthesis towards human preferences. Technically, beyond serving as an automatic VTON evaluator, VTON-VLLM upgrades VTON model through two pivotal ways: (1) providing fine-grained supervisory signals during the training of a plug-and-play VTON refinement model, and (2) enabling adaptive and preference-aware test-time scaling at inference. To benchmark VTON models more holistically, we introduce VITON-Bench, a challenging test suite of complex try-on scenarios, and human-preference–aware metrics. Extensive experiments demonstrate that powering VTON models with our VTON-VLLM markedly enhances alignment with human preferences. Code is publicly available at: https://github.com/HiDream-ai/VTON-VLLM/.

## 1  Introduction

Image-based Virtual Try-On (VTON) is an increasingly influential task in computer vision that aims to synthesize realistic images of a person wearing a specified garment, given the input images of both the individual and the clothing item. By allowing users to virtually preview apparel without the need for physical trials, VTON holds significant potential for transforming online shopping experiences and enabling new applications in fashion e-commerce.

Early GAN-based VTON methods [8, 11, 12, 13, 15, 16, 24, 42, 45] typically adopt a two-stage pipeline involving pose-guided garment warping and person-garment blending. These approaches often yield visually implausible garment details and struggle with complex human poses due to inaccurate warping process. Motivated by the success of diffusion models [2, 4, 6, 7, 17, 22, 33, 35, 51, 53], recent VTON approaches [9, 14, 29, 34, 41] leverage the pre-trained diffusion priors to enhance generation quality. These approaches generally incorporate garment-specific features through specialized modules (e.g., ControlNet [49] and ReferenceNet [46]), and synthesize high-fidelity images in a progressive manner.

Despite achieving low FID [18] and LPIPS [50] scores by these diffusion-based models, the resultant try-on images often fall short of fully meeting user expectations (e.g., the inconsistent garment details in the right example of Figure 1 (a)). These metrics are seemingly effective for measuring general image quality, but fail to adequately capture fine-grained visual attributes (e.g., texture preservation, and body–clothing coherence) that are critical for realistic and satisfactory virtual try-on experiences.

---

*This work was performed at HiDream.ai.

39th Conference on Neural Information Processing Systems (NeurIPS 2025).

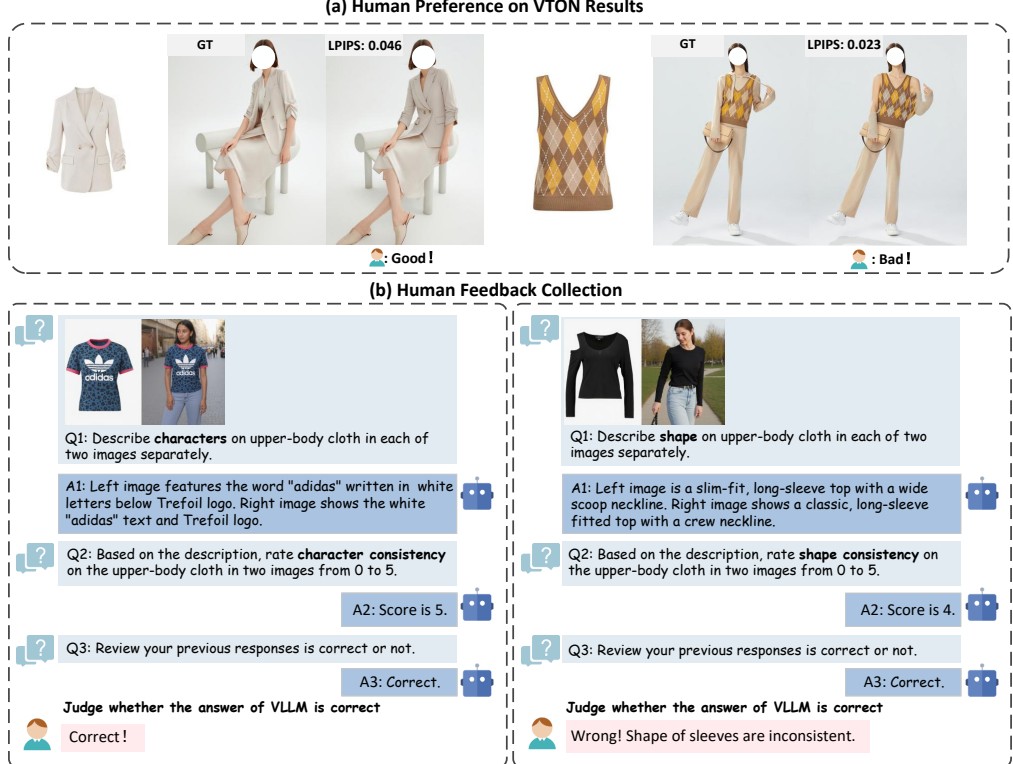

Figure 1: (a) Human Preference on VTON Results: Though low LPIPS scores are achieved, not all synthesized images generated by state-of-the-art VTON models fully meet user expectations. (b) Human Feedback Collection: A pre-trained VLLM is first leveraged to automatically assess the synthesized high-quality garment-person images step-by-step through multi-turn conversations. These assessments are subsequently reviewed and validated by human annotators to ensure reliability.

To mitigate these limitations, we introduce VTON-VLLM, a human-preference–aware vision large language model to both evaluate and steer VTON towards human preferences. We begin by collecting a new dataset of human feedback on VTON outputs (see Figure 1 (b)): each sample consists of a reference garment image $I_g$, a synthesized try-on image $I_h$, and fine-grained annotations across multiple perceptual criteria (e.g., visual pattern preservation, textual character correctness, and image quality). These annotations are organized into multi-turn conversational prompts for the fine-tuning of VTON-VLLM to reinforce chain-of-thought reasoning. Consequently, the learnt VTON-VLLM is able to meticulously analyze the visually grounded evidences in $I_h$ and score how well $I_h$ matches the reference garment $I_g$ as expected by humans step-by-step.

As a unified "fashion expert", VTON-VLLM then aligns VTON models with human preferences through two pivotal ways. (1) **Fine-grained supervision**: Given a low-quality output $I_h^{LQ}$ from any base VTON model, VTON-VLLM generates a detailed refinement instruction $t_r$ by comparing reference garment $I_g$ and $I_h^{LQ}$. Conditioned on such refinement instruction $t_r$, a plug-and-play VTON Refinement Model (VRM-Instruct) is trained to transform the low-quality output $I_h^{LQ}$ into its high-quality counterpart $I_h^{HQ}$. By doing so, any VTON models can be elegantly upgraded in a cascade manner without retraining primary VTON models. (2) **Adaptive and preference-aware test-time scaling**: At inference, VTON-VLLM further serves as a dynamic reward model, akin to test-time scaling in LLM [37], to iteratively steer VRM-Instruct's refinements based on the learnt human preferences. Such design adapts refinement strength on-the-fly, and ensures that each refinement incrementally improves VTON results.

In summary, our main contributions are threefold:

• We present VTON-VLLM, a reliable human-preference-aligned evaluator, which is learnt by tuning VLLM with our newly collected human feedback on synthesized VTON data. Our VTON-VLLM

goes beyond typical perceptual metrics and aligns better with human preference in image quality assessment tailored for VTON.

- By taking VTON-VLLM as a unified fashion expert, we further trigger human-preference-driven enhancement in any VTON models via two ways: 1) providing fine-grained supervision for training a plug-and-play VTON refinement model that elevates low-quality outputs to high-quality standards, and 2) triggering adaptive and preference-aware test-time scaling to iteratively guide VTON refinements at inference.

- We construct a newly collected test set that features challenging try-on scenarios, dubbed as VITON-Bench, alongside novel human-preference-aware metrics powered by VTON-VLLM. This benchmark enables more holistic assessment of VTON models, ensuring that quantitative evaluations align with real user expectations well.

## 2   Related Work

**Virtual Try-on.** Early virtual try-on (VTON) methods [15, 13, 42, 45] mainly capitalize on a two-stage pipeline: first, deforming the garment to match the target body shape, and then synthesizing the final image using a GAN-based generator, conditioned on the warped garment and human pose. Most recently, advanced approaches have explored the integration of powerful diffusion generation models into VTON pipelines to better preserve garment appearance and achieve realism. [14, 29] applies a straightforward approach by overlaying the warped garment on the target person to guide the diffusion model. However, these methods heavily rely on deformed garments and thus easily introduce artifacts due to imperfect warping. To address this, Stable-VTON [34] extends ControlNet [49] to learn implicit garment deformation. OOTDiffusion [46] and IDM-VTON [9] adopt a dual-UNet framework to better leverage pre-trained image priors within diffusion models for representation learning of garment. Furthermore, SPM-Diff [40] incorporates the visual correspondence between the garment and the human body into the dual-UNet architecture, which enhances the preservation of fine-grained details and clothing outline. Although promising results are achieved, these diffusion-based VTON models still fail to generate images that fully meet user expectations.

**Human Preference Alignment in VLLMs.** In recent years, vision large language models (VLLMs) have exhibited remarkable performance in tasks involving cross-modal understanding [28, 52], reasoning [5, 32], and interaction [19, 48]. However, the modality gap between vision and language leads to significant vision hallucinations in VLLMs, wherein the generated text is inconsistent with the associated images. Inspired by the reinforcement learning from human preference in large language models [31, 38], advanced works [39, 47] propose collecting human feedback on vision hallucinations, such as visual instructions or error corrections, to address the issue of cross-modal misalignment. However, integrating human preference alignment into VTON remains largely unexplored.

## 3   VTON-VLLM: Human Preference-Aware VLLM for VTON

In this section, we first introduce VTON-VLLM, a human-preference-aware vision large language model (VLLM) for virtual try-on (VTON). Given a reference garment image and a synthesized person image wearing this garment, the proposed VTON-VLLM can comprehensively assess whether the synthesized image aligns with human preferences in both image quality and detail preservation.

### 3.1   Motivation

The recent advancements in VTON techniques have significantly improved the quality of synthesized images, enabling practical applications in fashion e-commerce and digital styling. Although promising results are achieved, existing methods still fail to generate images that are fully aligned with human preferences. For example, as shown in Figure 1 (a), inconsistencies and incoherence often arise in detail preservation and body alignment, respectively. And these types of defects may not be effectively captured by traditional metrics such as the Learned Perceptual Image Patch Similarity (LPIPS) and Frechet Inception Distance (FID) in one-to-many image synthesis. Thus, it is essential to accurately evaluate whether a synthesized image fully aligns with human preferences.

Recent works [25, 44] have explored using vision large language model (VLLM) as proxies for human judgment, leveraging their powerful capabilities in multi-modal understanding. However, the efficacy of these methods in evaluating the quality of VTON image remains unsatisfactory, particularly when

synthesized fashion images exhibit inconsistent fine-grained attributes with the reference garment. To address this issue, we develop a human-preference-aware VLLM by first collecting comprehensive human feedback on synthesized VTON data across multiple perceptual criteria in VTON, which are then organized into multi-turn conversational prompts. These data are further used to fine-tune a pre-trained VLLM, facilitating chain-of-thought reasoning for quality assessment.

## 3.2 Human Feedback Dataset

To better investigate human judgments in VTON, we establish an evaluation framework that includes garment consistency and image quality. The two metrics encompass multiple sub-aspects as follows:

**Garment Consistency (GC).** This criterion evaluates whether the generated deformed garment retains the original 1) *visual pattern* (i.e., texture and logo), 2) *text characters*, 3) *sleeve style* (e.g., wave or puff sleeves) and 4) *holistic garment shape*.

**Image Quality (IQ).** This criterion assesses 1) *edge artifacts* near garment-body intersections and 2) *pose plausibility* of the target person in the synthesized image.

When constructing the human feedback dataset, we first employ a pre-trained VLLM to roughly annotate 40K synthesized images according to the pre-defined evaluation specifications. 15 annotators from different educational backgrounds are then invited to review and validate these annotations. As a result, we obtain a high-quality dataset consisting of clothing-model pairs, fine-grained human feedback and quality scores across the above six fine-grained sub-aspects. A multi-turn conversational prompt $[X_q^1, X_a^1, \cdots, X_q^M, X_a^M]$ is then constructed for each pair, where $X_q^*$ and $X_a^*$ denote a question string and an answer string, respectively. This way, the trained model is capable of decomposing the complex evaluation into intermediate reasoning steps (i.e., chain-of-thought reasoning) for improved VTON understanding, based on which faithful quality scores are computed.

## 3.3 Training

We adapt a pre-trained VLLM [1] to comprehensively assess image quality in alignment with human judgments for VTON by minimizing the cross-entropy loss on the collected human feedback dataset. Specifically, we form all the multi-turn conversational instruction strings into a single input sequence with $L$ tokens. Hence, given an input instruction $X_q$, we compute the probability of $X_a$ as:

$$p(X_a|I_v, X_q) = \prod_{i=1}^{L} p_\theta(x_i|I_v, X_{q,<i}, X_{a,<i})$$

where $\theta$ represents the trainable parameters, and $I_v$ is the concatenated clothing-model image. Let $X_{q,<i}$ and $X_{a,<i}$ denote the historical instructions and answer tokens in all turns before the current prediction token $x_i$, respectively.

# 4 Align VTON models with Human Preferences via VTON-VLLM

This section details how VTON-VLLM aligns VTON models with human preferences by: 1) providing fine-grained supervision for a plug-and-play VTON refinement model (Section 4.1), and 2) facilitating test-time scaling in inference with human-preference-aware reward (Section 4.2).

## 4.1 VTON Refinement Model with Fine-grained Supervision

Though existing VTON approaches achieve strong quantitative results on standard benchmarks, their generated outputs often fail to meet the user expectations. To address this limitation, we introduce a VTON refinement model optimized with fine-grained supervision from the VTON-VLLM, dubbed as VRM-Instruct. The refinement model can be cascaded with arbitrary VTON models in a plug-and-play manner, consistently improving their generation quality and alignment with human preferences.

Recall that our proposed VTON-VLLM can not only score a garment-person pair from six fine-grained aspects but also excel in reasoning for interpretable visual evidences to justify the scoring decisions in Section 3. Taking advantage of this capability, we first construct triplets $(I_g, I_h^{LQ}, I_h^{HQ})$ consisting of a reference garment $I_g$, a low-quality person image $I_h^{LQ}$ and a high-quality version

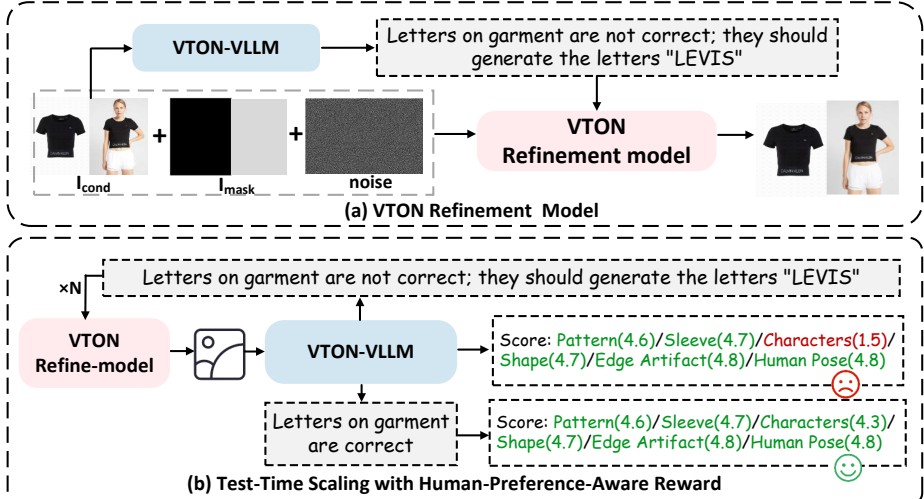

Figure 2: Illustration of our proposed VTON refinement model (VRM-Instruct) and test-time scaling (TTS), both of which are powered by our VTON-VLLM. (a) Given a sample generated by any VTON model, VTON-VLLM is first employed to evaluate the synthesized image via chain-of-thought reasoning and then provide a fine-grained instruction to guide VRM-Instruct for generation of better results that meet user expectations. (b) TTS can be readily triggered by collaborating VTON-VLLM and VRM-Instruct, further iteratively enhancing the outputs to align human preferences.

$I_h^{HQ}$. For each triplet, our VTON-VLLM generates a fine-grained refinement instruction $t_r$ that pinpoints specific regions in $I_h^{LQ}$ that do not align human preferences in either garment consistency or image quality by comparing $I_g$ and $I_h^{LQ}$. Then, VRM-Instruct is steered to enhance $I_h^{LQ}$ into $I_h^{HQ}$ following the instruction. In this work, $I_h^{HQ}$ is sampled from high-quality datasets (e.g., VITON-HD) while $I_h^{LQ}$ is synthesized by a less capable VTON model. Inspired by [21, 10], we propose incorporating the intrinsic in-context visual priors of a pre-trained text-to-image diffusion transformer (i.e., FLUX-Fill [2]) into our VRM-Instruct, which frames VTON synthesis as conditional image inpainting. Specifically, we concatenate the garment image $I_g \in \mathbb{R}^{H \times W \times 3}$ with the low-quality image $I_h^{LQ} \in \mathbb{R}^{H \times W \times 3}$, resulting in visual condition $I_{cond} = I_g \copyright I_h^{LQ}$. Meanwhile, the ground-truth image is defined as $I_{gt} = I_g \copyright I_h^{HQ}$. The inpainting mask $I_{mask} = I_0 \copyright I_1$ is derived by combining a black image $I_0$ and a white image $I_1$, indicating the untouched area and the inpainting region, respectively. The objective function for training VRM-Instruct can be formulated as:

$$\mathcal{L}_{DiT} = E_{t, \epsilon \sim \mathcal{N}(0, I)} ||v_\gamma([\mathbf{x}_t, \mathbf{x}_{cond}, \mathbf{x}_{mask}], t_r) - (\epsilon - \mathbf{x}_0)||_2^2, \tag{1}$$

where $\mathbf{x}_0$, $\mathbf{x}_{cond}$ and $\mathbf{x}_{mask}$ are the latent codes of $I_{gt}$, $I_{cond}$ and $I_{mask}$, respectively. $\mathbf{x}_t$ is obtained by adding Gaussian noises $\epsilon$ to the latent code $\mathbf{x}_0$ at timestep $t$. $\gamma$ denotes the trainable parameters.

At inference time, our VTON-VLLM is first utilized to identify perceptually defective regions in the initial output from an arbitrary VTON model and generate the refinement instruction $t_r$. The low-quality initial image and the instruction are then fed into our VRM-Instruct to produce a high-fidelity image that better aligns with learnt human preferences, as illustrated in Figure 2 (a).

## 4.2 Test-Time Scaling with Human Preference-Aware Reward

Test-time scaling (TTS) has demonstrated significant effectiveness in both linguistic and multi-modal understanding tasks [7, 37]. By further investigating our VTON-VLLM and VRM-Instruct, we found that TTS is naturally supported through the mutual collaboration of these two models. Specifically, following the prediction of VRM-Instruct, VTON-VLLM functions as a human-preference-aware reward model to assess the quality of the refined image. Then, another forward pass of VRM-Instruct is executed if this image does not fully align with human preferences, conditioned on the refined image and the updated refinement instruction, yielding an incrementally improved output. This

---

[2]https://huggingface.co/black-forest-labs/FLUX.1-Fill-dev

iterative process continues until the output image meets the pre-defined quality criteria or a maximum of $N$ iterations is reached. TTS for VTON is shown in Figure 2 (b).

# 5 Experiments

## 5.1 Experimental Setups

**Datasets.** VITON-HD [8] contains 13,679 frontal-view image pairs of women and upper garments. Following previous works [14, 29], we split the dataset into 11,647 training pairs and 2,032 testing pairs. DressCode [30] consists of 53,795 image pairs, divided into three categories: 15,366 upper-body clothes, 8,951 lower-body clothes, and 29,478 dresses. We adopt the official split, using 1,800 pairs from each category for testing and the remaining pairs for training.

**Evaluation.** Generally, two evaluation settings are adopted in VTON: paired and unpaired. Specifically, the paired setting involves reconstructing the ground-truth person image using the original garment, while the unpaired setting requires replacing the clothing of a person image with a different garment. To benchmark VTON models more holistically, we newly curate a challenging test set that features realistic and complex VTON scenarios, namely VITON-Bench. Particularly, VITON-Bench consists of 1,000 paired garment-person images and 2,300 unpaired images manually collected from the Internet, VITON-HD and CosmicMan [26]. All the methods are evaluated on three test sets: VITON-HD, DressCode, and our VITON-Bench. For the paired setting, SSIM [43] and LPIPS [50] are commonly adopted to measure visual similarity between the generated images and the ground-truth images. Furthermore, we leverage our proposed VTON-VLLM as a "fashion expert" to compute human-preference-aware metrics, which evaluate both garment consistency (GC) and image quality (IQ) across a set of fine-grained attributes, including visual patterns, text characters, sleeve style, garment shape, edge artifacts, and human pose. For the unpaired setting where ground-truth references are unavailable, we employ FID [18], KID [3], GC and IQ to assess generation quality. It is worth noting that our GC metric can evaluate the visual consistency between the synthesized images and input garments in the unpaired setting, while FID and KID primarily measure the perceptual realism and distribution similarity of the generated images with respect to a target dataset.

**Implementation Details.** (1) VTON-VLLM: Our VTON-VLLM is initialized from Pixtral12B [1] and further fine-tuned on the collected human feedback dataset. The model is trained for 2 epochs with a batch size of 64. The learning rate is set to 0.0001, and AdamW [27] is employed as the optimizer. We incorporate Low-Rank Adaptation (LoRA) [20] with a rank of 16 for training efficiency. (2) VTON Refinement Model: We also employ AdamW to optimize the model over 50,000 training steps. The learning rate is set to 0.00005 with a warmup over 500 iterations, and the batch size is 1.

## 5.2 Results

**Quantitative Results on VITON-HD and DressCode.** Table 1 presents the comparisons among different methods on VITON-HD. Note that $GC_p/IQ_p$ and $GC_u/IQ_u$ are human-preference-aware metrics driven by our VTON-VLLM for paired and unpaired settings, respectively. Although state-of-the-art models such as IDM-VTON and SPM-Diff have already achieved promising results, further improvements are realized through our VRM-Instruct and TTS module, both of which are powered by VTON-VLLM. Specifically, in the paired setting, **IDM-VTON w/ VRM-Instruct and TTS** not only significantly reduces LPIPS from 0.082 to 0.051 but also increases $GC_p$ from 4.665 to 4.873, demonstrating better visual consistency compared to its baseline counterpart **IDM-VTON**. Moreover, in the unpaired setting, **IDM-VTON w/ VRM-Instruct and TTS** demonstrates consistent performance gains, as evidenced by a decrease in KID from 0.741 to 0.634 and an increase in $IQ_u$ from 4.392 to 4.711, yielding absolute improvements of 0.107 and 0.319, respectively. These observations highlight the advantages of leveraging our VRM-Instruct and TTS to steer VTON models towards human preferences in both image quality and garment consistency, which is achieved through iterative refinement guided by fine-grained and preference-aligned instructions from VTON-VLLM. Please refer to the Appendix for more results on DressCode.

**Quantitative Results on VITON-Bench.** We quantitatively evaluate different VTON models on the newly introduced VITON-Bench, and Table 2 summarizes the results. The increased complexity of VITON-Bench leads to substantial performance degradation in most metrics, with particularly pronounced drops in SSIM, FID, and $IQ_*$. Conversely, $GC_*$ exhibits relatively mild decreases. Upon

Table 1: Quantitative results on VITON-HD. $GC_*$ and $IQ_*$ denotes the proposed human-preference-aware metrics Garment Consistency and Image Quality, respectively. VRM-Instruct and TTS are short for VTON refinement model with fine-grained instructions and test-time scaling, respectively.

| Train/Test | VITON-HD/VITON-HD | | | | Average Preference-Aware Scores | | | |
|---|---|---|---|---|---|---|---|---|
| Method | SSIM ↑ | LPIPS ↓ | FID ↓ | KID ↓ | $GC_p$ ↑ | $IQ_p$ ↑ | $GC_u$ ↑ | $IQ_u$ ↑ |
| LaDI-VTON [29] | 0.864 | 0.096 | 10.531 | 2.303 | 4.542 | 4.621 | 4.320 | 4.291 |
| Stable-VTON [23] | 0.852 | 0.084 | 10.200 | 1.921 | 4.625 | 4.660 | 4.331 | 4.405 |
| OOTDiffusion [46] | 0.881 | 0.071 | 10.153 | 0.892 | 4.619 | 4.649 | 4.400 | 4.388 |
| IDM-VTON [9] | 0.877 | 0.082 | 9.372 | 0.741 | 4.665 | 4.750 | 4.445 | 4.392 |
| w/ VRM-Instruct | 0.880 | 0.054 | 9.012 | 0.667 | 4.870 | 4.852 | 4.712 | 4.677 |
| w/ VRM-Instruct and TTS | **0.882** | **0.051** | **8.902** | **0.634** | **4.873** | **4.920** | **4.724** | **4.711** |
| SPM-Diff [40] | **0.911** | 0.063 | 9.034 | 0.720 | 4.742 | 4.775 | 4.502 | 4.411 |
| w/ VRM-Instruct | 0.890 | **0.054** | 9.012 | 0.667 | 4.891 | 4.850 | 4.715 | 4.680 |
| w/ VRM-Instruct and TTS | 0.892 | 0.055 | **9.010** | **0.663** | **4.900** | **4.930** | **4.745** | **4.725** |
| CAT-VTON [10] | 0.870 | 0.057 | 9.015 | 0.670 | 4.713 | 4.765 | 4.510 | 4.420 |
| w/ VRM-Instruct | 0.885 | 0.055 | 8.915 | 0.660 | 4.832 | 4.834 | 4.673 | 4.661 |
| w/ VRM-Instruct and TTS | **0.891** | **0.052** | **8.923** | **0.652** | **4.895** | **4.900** | **4.733** | **4.738** |

Table 2: Quantitative results on the newly introduced VITON-Bench. $GC_*$ and $IQ_*$ denotes the proposed human-preference-aware metrics Garment Consistency and Image Quality, respectively.

| Train/Test | VITON-HD/VITON-Bench | | | | Average Preference-Aware Scores | | | |
|---|---|---|---|---|---|---|---|---|
| Method | SSIM ↑ | LPIPS ↓ | FID ↓ | KID ↓ | $GC_p$ ↑ | $IQ_p$ ↑ | $GC_u$ ↑ | $IQ_u$ ↑ |
| LaDI-VTON [29] | 0.702 | 0.173 | 14.032 | 2.03 | 4.107 | 3.692 | 3.631 | 3.021 |
| Stable-VTON [23] | 0.725 | 0.110 | 13.566 | 1.67 | 4.280 | 4.010 | 3.894 | 3.083 |
| OOTDiffusion [46] | 0.733 | 0.097 | 13.087 | 1.43 | 4.357 | 4.036 | 3.926 | 3.447 |
| IDM-VTON [9] | 0.750 | 0.083 | 12.562 | 1.02 | 4.374 | 4.069 | 4.026 | 3.825 |
| w/ VRM-Instruct | 0.787 | 0.064 | 10.931 | 0.92 | 4.566 | 4.398 | 4.341 | 4.126 |
| w/ VRM-Instruct and TTS | **0.794** | **0.060** | **10.784** | **0.87** | **4.612** | **4.429** | **4.442** | **4.302** |
| SPM-Diff [40] | 0.765 | 0.084 | 12.208 | 1.34 | 4.519 | 4.230 | 4.235 | 3.804 |
| w/ VRM-Instruct | 0.782 | 0.069 | 11.011 | 0.94 | 4.570 | 4.401 | 4.288 | 4.027 |
| w/ VRM-Instruct and TTS | **0.797** | **0.062** | **10.627** | **0.87** | **4.623** | **4.429** | **4.407** | **4.280** |
| CAT-VTON [10] | 0.751 | 0.078 | 12.663 | 1.23 | 4.518 | 4.238 | 4.150 | 3.991 |
| w/VRM-Instruct | 0.772 | 0.063 | 10.721 | 0.90 | 4.542 | 4.579 | 4.262 | 4.156 |
| w/ VRM-Instruct and TTS | **0.788** | **0.060** | **10.232** | **0.85** | **4.555** | **4.685** | **4.390** | **4.353** |

examining the synthesized results, we observed that these state-of-the-art VTON models are capable of preserving parts of the visual characteristics in the garment by learning robust garment features. However, they struggle with handling garment warping for complex human poses in real-world scenarios, resulting in notable artifacts and incorrect fine-grained details. Despite these challenges, the integration of our proposed VRM-Instruct and TTS further enhances the performance of all the methods. Particularly, **SPM-Diff w/ VRM-Instruct and TTS** outperforms the baseline **SPM-Diff**, achieving improvements in both $GC_u$ (from 4.235 to 4.407) and $IQ_u$ (from 3.804 to 4.280). These results clearly demonstrate that VRM-Instruct and TTS can effectively improve synthesized outputs, even in intricate real-world cases, by capitalizing on human preferences learnt by VTON-VLLM.

**Qualitative Results.** We conducted a qualitative evaluation to compare the results synthesized by different methods on VITON-Bench. Some examples are illustrated in Figure 3. It can be observed that our proposed VRM-Instruct and TTS module consistently improve the state-of-the-art baselines, i.e., LaDI-VTON, IDM-VTON and CAT-VTON. For instance, in the first row, the enhanced images exhibit significantly better visual quality compared to those generated by the baseline models in the challenging case with complex human poses. Moreover, in the second row, the images synthesized by baseline models suffer blurred or repeated text rendering and severe edge artifacts, whereas the counterparts refined by incorporating our VRM-Instruct and TTS achieve superior text fidelity. This underscores the effectiveness of our proposed designs (i.e., VRM-Instruct and TTS), which iteratively enhance the synthesized images with fine-grained and human-preference-aligned instructions from VTON-VLLM. Please refer to the Appendix for more examples.

### 5.3 Discussions

**Effect of VTON-VLLM in VTON Synthesis.** As previously noted, our VTON-VLLM can upgrade arbitrary VTON models in a cascade manner by employing a plug-and-play VTON refinement model

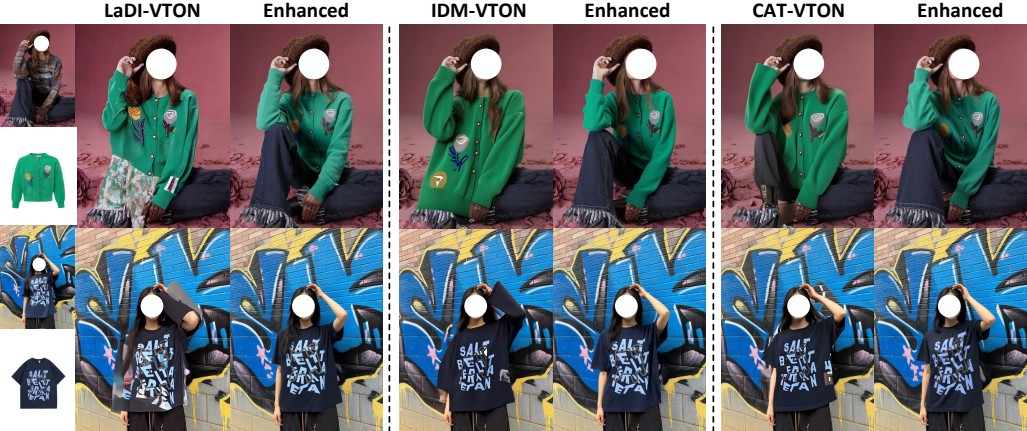

Figure 3: Qualitative results on VITON-Bench. The first and second rows show results under unpaired and paired settings, respectively. For each pair, the left image is generated by a baseline model (i.e., LaDI-VTON, IDM-VTON, CAT-VTON), and the right image is enhanced by incorporating our proposed VRM-Instruct and TTS module powered by VTON-VLLM.

Table 3: Ablation study of VRM-Instruct and TTS module on VITON-HD. VRM-Instruct denotes the proposed VTON refinement model with fine-grained instructions from VTON-VLLM. TTS denotes test-time scaling.

| Model | Garment Consistency | | | | Image Quality | | | |
|---|---|---|---|---|---|---|---|---|
| | Pattern | Characters | Sleeve | Shape | Edge Artifact | Human Pose | SSIM ↑ | LPIPS ↓ |
| **Base** | 4.645 | 4.698 | 4.679 | 4.833 | 4.650 | 4.821 | 0.870 | 0.057 |
| **Base + VRM-Instruct** | 4.873 | 4.800 | 4.782 | 4.865 | 4.803 | 4.865 | 0.885 | 0.055 |
| **Base + VRM-Instruct + TTS** | **4.931** | **4.913** | **4.865** | **4.870** | **4.889** | **4.913** | **0.891** | **0.052** |

that utilizes fine-grained refinement instructions (**VRM-Instruct**), and further function as a human-preference-aware reward model to iteratively guide VRM-Instruct based on learnt human preferences, akin to test-time scaling (**TTS**). In this part, we conduct an ablation study to evaluate individual contributions of these two core designs by sequentially adding VRM-Instruct and TTS to a **Base** VTON model. The scores across all six fine-grained dimensions of garment consistency (GC) and image quality (IQ) are shown. CAT-VTON is adopted as the Base model here. As shown in Table 3, by incorporating the fine-grained instructions from VTON-VLLM to identify imperfect regions in the garment and guide the refinement process towards human preferences, **Base + VRM-Instruct** achieves notable improvements over **Base** model in garment consistency. Specifically, compared with **Base** model, **Base + VRM-Instruct** boosts "Characters" and "Pattern" from 4.698 to 4.800 and from 4.645 to 4.873, respectively. Furthermore, we integrate TTS into **Base + VRM-Instruct** to enable iterative enhancement with human-preference-aware reward provided by VTON-VLLM in the full run **Base + VRM-Instruct + TTS**, surpassing all the other settings. Compared to **Base** model, the "Edge Artifact" and SSIM scores increase from 4.650 to 4.889 and from 0.870 to 0.891, respectively.

**Preference Agreement.** In this part, we evaluate the alignment between human preferences and our VTON-VLLM. Specifically, we adopt a held-out set from our collected human feedback data as ground-truth annotations, and compute the percentage of consistent judgements between human and VTON-VLLM as a measure of inter-rater agreement. Chain-of-thought reasoning is employed in VTON-VLLM to generate the fine-grained assessments of the synthesized images. Our VTON-VLLM attains an agreement rate of 84.52%, demonstrating strong alignment with human preferences.

**FID vs VTON-VLLM Score.** In Table 4, we further investigate the correlation between the no-reference evaluation metric FID/VTON-VLLM and human judgments by dividing images from the held-out human feedback data into subsets based on whether the generated images are preferred by humans or not. FID and the averaged VTON-VLLM scores are calculated for each subset. Note that no-reference evaluation refers to image quality assessment performed in the absence of ground-truth images. As shown, there is no significant difference in FID scores between the two subsets. This suggests that FID may not effectively capture the subjective dimensions underlying human judgments. In contrast, VTON-VLLM yields higher scores for the human-preferred subset compared to the non-preferred one, confirming its better consistency with human perceptual criteria.

Table 4: Analysis of the alignment between our VTON-VLLM and human preferences. (1) FID and VTON-VLLM scores are calculated for subsets of preferred and non-preferred images by human. (2) Preference agreement refers to the inter-rater agreement between human and VTON-VLLM.

| Method | Preferred | Non-preferred | Preference Agreement |
|---|---|---|---|
| **FID** | 6.84 | 6.99 | / |
| **VTON-VLLM** | 4.92 | 3.94 | 84.52% |

Table 5: Analysis of our VTON-VLLM in high-quality dataset construction. VITON-SYN is a synthesized dataset by leveraging a VTON model for generation and VTON-VLLM for data filtering.

| Training Data | Garment Consistency | | | | Image Quality | | SSIM ↑ | LPIPS ↓ |
|---|---|---|---|---|---|---|---|---|
| | Pattern | Characters | Sleeve | Shape | Edge Artifact | Human Pose | | |
| **VITON-HD** | 4.861 | 4.724 | 4.734 | 4.676 | 4.714 | 4.588 | 0.875 | 0.045 |
| **VITON-SYN** | 4.907 | 4.868 | 4.782 | 4.832 | 4.856 | 4.672 | 0.898 | 0.044 |
| **VITON-HD + VITON-SYN** | **4.931** | **4.912** | **4.822** | **4.872** | **4.890** | **4.726** | **0.903** | **0.043** |

**High-quality Dataset Construction with VTON-VLLM.** We conduct an ablation study to explore the intrinsic ability of our VTON-VLLM as an image quality evaluator in high-quality dataset construction. Specifically, we first train a VTON model that is architecturally similar to VRM-Instruct on the VITON-HD dataset, where the garment image $I_g$ and a blank white image $I_1$ are combined as the visual condition $I_{cond}$ during training. More implementation details are provided in Appendix Sec A.2. The trained model is then employed to synthesize garment–person pairs, using garment images sampled from existing datasets [8, 30, 36] and textual prompts describing diverse scenes and poses from a predefined pool. Finally, the generated images undergo rigorous data filtering by our VTON-VLLM, resulting in a human-preference-aligned training dataset **VITON-SYN**. In Table 5, we compare our VTON models trained on VITON-HD, VITON-SYN, and the combination of both. Surprisingly, training the VTON model solely on VITON-SYN outperforms the in-domain run **VITON-HD**. We attribute this improvement to two key factors: (1) VITON-SYN exhibits comparable visual quality to the manually collected VITON-HD and scalability in data quantity, and (2) the VTON model has gained enhanced capability in handling diverse scenarios when trained on VITON-SYN. This demonstrates the effectiveness of VTON-VLLM in performing quality assessment for the construction of high-quality datasets. Further improvements in all metrics are achieved by leveraging both sets for training. Please refer to the Appendix for details of VITON-SYN.

## 6 Conclusion

In this paper, we have presented VTON-VLLM, a vision large language model fine-tuned with human feedback on synthesized VTON data, which can comprehensively assess whether synthesized images meet user expectations. As a unified "fashion expert", our VTON-VLLM can elegantly upgrade VTON models and align them with human preferences through two pivotal ways: 1) providing fine-grained supervision for the training of a plug-and-play VTON refinement model, and 2) enabling test-time scaling with human-preference-aware reward for adaptive inference. To benchmark VTON models more holistically, we curate a challenging test set (i.e., VITON-Bench), featuring realistic and complex VTON scenarios, and introduce human-preference-aware evaluation metrics powered by our VTON-VLLM. Extensive experiments on VITON-HD, DressCode, VITON-Bench validate that incorporating our VTON-VLLM leads to significant improvements in VTON synthesis.

**Limitations.** Despite the above advantages, our method still has several limitations. First, it may encounter difficulties in handling rare or long-tail clothing items with distinctive patterns or structures. Second, objects held by the person and significantly overlapping with the target garment regions may not be accurately reconstructed, as our method primarily emphasizes garment preservation. Finally, similar to test-time scaling techniques adopted in large language models, our method necessitates additional computational overhead during the inference.

**Border Impacts.** Similar to popular generative models for visual content creation, our proposed designs allow anyone to produce high-fidelity person images, which holds great potential impact to revolutionize the shopping experience in the e-commerce industry. However, these synthesized person images risk exacerbating issues of misinformation and deepfakes. To mitigate the misuse of our models, we will enforce strict adherence to usage guidelines and integrate an additional pipeline to automatically embed digital watermarks into the synthesized outputs.

**Acknowledgement.** This work was supported in part by the Beijing Municipal Science and Technology Project No. Z241100001324002 and Beijing Nova Program No. 20240484681.

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

# A Appendix

## A.1 More Quantitative and Qualitative Results

**Quantitative Results on DressCode.** The quantitative results on DressCode across three macro-categories are summarized in Table 6 and Table 7. Table 6 reports SSIM, LPIPS, $GC_p$, and $IQ_p$ under the *paired* setting, while Table 7 presents results under the *unpaired* setting, including FID, KID, $GC_u$, and $IQ_u$. With the integration of our proposed VRM-Instruct and TTS modules, state-of-the-art VTON models achieve further improvements on the DressCode benchmark. Specifically, in the paired setting for the upper-body category, **OOTDiffusion w/ VRM-Instruct + TTS** increases SSIM from 0.902 to 0.928 and $GC_p$ from 3.59 to 4.14, indicating enhanced detail preservation. Meanwhile, in the unpaired setting, the same recipe boosts $IQ_u$ from 4.05 to 4.15 and reduces FID from 11.03 to 9.43, which demonstrates improved visual realism. These results on the DressCode dataset further validate the effectiveness of our VRM-Instruct and TTS modules in guiding VTON models towards better alignment with human preferences in both image quality and garment fidelity.

**More Qualitative Results.** In Figure 4, we present additional qualitative comparisons of synthesized results produced by different methods on VITON-HD. Furthermore, more results for other clothing categories (i.e., lower-body garments and dresses) on DressCode are presented in Figure 5.

Table 6: Quantitative results on DressCode in paired setting of three categories(D.C.Upper, D.C.Lower, D.C.Dress). VRM-Instruct and TTS are short for VTON refinement model with fine-grained instructions and test-time scaling, respectively.

| Train/Test | D.C.Upper/D.C.Upper | | | | D.C.Lower/D.C.Lower | | | | D.C.Dress/D.C.Dress | | | |
|---|---|---|---|---|---|---|---|---|---|---|---|---|
| Method | SSIM ↑ | LPIPS ↓ | $GC_p$ ↑ | $IQ_p$ ↑ | SSIM ↑ | LPIPS ↓ | $GC_p$ ↑ | $IQ_p$ ↑ | SSIM ↑ | LPIPS ↓ | $GC_p$ ↑ | $IQ_p$ ↑ |
| **LaDI-VTON** [29] | 0.904 | 0.063 | 3.48 | 3.32 | 0.862 | 0.122 | 3.36 | 3.22 | 0.782 | 0.129 | 3.25 | 3.02 |
| w/ **VRM-Instruct** and **TTS** | **0.930** | **0.042** | **4.12** | **3.99** | **0.923** | **0.043** | **4.06** | **3.81** | **0.912** | **0.069** | **3.97** | **3.76** |
| **OOTDiffusion** [46] | 0.902 | 0.050 | 3.59 | 3.43 | 0.882 | 0.103 | 3.42 | 3.26 | 0.824 | 0.100 | 3.40 | 3.15 |
| w/ **VRM-Instruct** and **TTS** | **0.928** | **0.039** | **4.14** | **3.92** | **0.918** | **0.044** | **4.01** | **3.76** | **0.898** | **0.064** | **3.92** | **3.74** |
| **IDM-VTON** [9] | 0.911 | 0.060 | 3.70 | 3.57 | 0.913 | 0.055 | 3.60 | 3.33 | 0.863 | 0.082 | 3.62 | 3.44 |
| w/ **VRM-Instruct** and **TTS** | **0.931** | **0.034** | **4.08** | **3.98** | **0.917** | **0.040** | **3.92** | **3.72** | **0.883** | **0.062** | **3.94** | **3.72** |
| **SPM-Diff** [40] | 0.927 | 0.042 | 3.74 | 3.60 | 0.914 | 0.050 | 3.70 | 3.45 | **0.892** | 0.073 | 3.68 | 3.49 |
| w/ **VRM-Instruct** and **TTS** | **0.928** | **0.035** | **4.06** | **4.03** | **0.920** | **0.046** | **4.03** | **3.79** | 0.890 | **0.069** | **3.91** | **3.70** |
| **CAT-VTON** [10] | 0.919 | 0.045 | 3.68 | 3.58 | 0.897 | 0.063 | 3.65 | 3.40 | **0.900** | **0.065** | 3.54 | 3.41 |
| w/ **VRM-Instruct** and **TTS** | **0.927** | **0.039** | **4.11** | **4.00** | **0.918** | **0.042** | **3.99** | **3.74** | 0.892 | **0.065** | **3.93** | **3.76** |

Table 7: Quantitative results on DressCode in unpaired setting of three categories(D.C.Upper, D.C.Lower, D.C.Dress). VRM-Instruct and TTS are short for VTON refinement model with fine-grained instructions and test-time scaling, respectively.

| Train/Test | D.C.Upper/D.C.Upper | | | | D.C.Lower/D.C.Lower | | | | D.C.Dress/D.C.Dress | | | |
|---|---|---|---|---|---|---|---|---|---|---|---|---|
| Method | FID ↓ | KID ↓ | $GC_u$ ↑ | $IQ_u$ ↑ | FID ↓ | KID ↓ | $GC_u$ ↑ | $IQ_u$ ↑ | FID ↓ | KID ↓ | $GC_u$ ↑ | $IQ_u$ ↑ |
| **LaDI-VTON** [29] | 14.26 | 3.33 | 4.02 | 3.78 | 13.38 | 1.98 | 2.86 | 2.64 | 13.12 | 1.85 | 3.21 | 2.99 |
| w/ **VRM-Instruct** and **TTS** | **9.38** | **0.52** | **4.34** | **4.19** | **9.80** | **0.85** | **3.27** | **3.10** | **10.65** | **0.84** | **3.40** | **3.55** |
| **OOTDiffusion** [46] | 11.03 | 0.86 | 4.10 | 4.05 | 9.62 | 0.84 | 2.99 | 2.92 | 10.65 | 0.84 | 3.40 | 3.29 |
| w/ **VRM-Instruct** and **TTS** | **9.43** | **0.42** | **4.31** | **4.15** | **9.66** | **0.80** | **3.19** | **3.08** | **9.69** | **0.52** | **3.76** | **3.58** |
| **IDM-VTON** [9] | 10.86 | 0.62 | 4.13 | 4.11 | 12.05 | 0.93 | 3.08 | 3.02 | 12.33 | 1.41 | 3.48 | 3.29 |
| w/ **VRM-Instruct** and **TTS** | **9.55** | **0.43** | **4.30** | **4.18** | **9.74** | **0.77** | **3.28** | **3.03** | **9.92** | **0.54** | **3.80** | **3.51** |
| **SPM-Diff** [40] | 10.56 | **0.40** | 4.23 | 4.15 | **9.02** | 0.80 | 3.11 | 3.02 | 10.17 | 0.50 | 3.55 | 3.49 |
| w/ **VRM-Instruct** and **TTS** | **9.46** | 0.45 | **4.32** | **4.20** | 9.18 | **0.78** | **3.26** | **3.11** | **9.79** | **0.51** | **3.78** | **3.54** |
| **CAT-VTON** [10] | **8.92** | 0.51 | 4.16 | 4.12 | 9.21 | 0.94 | 3.06 | 2.98 | 9.76 | 0.66 | 3.50 | 3.41 |
| w/ **VRM-Instruct** and **TTS** | 9.23 | **0.44** | **4.35** | **4.14** | **9.20** | **0.76** | **3.19** | **3.12** | **9.68** | **0.57** | **3.83** | **3.61** |

## A.2 More Discussions

**Effect of Iteration Count $N$ in Test-Time Scaling.** We analyze the sensitivity of our proposed TTS with respect to the iteration count $N$, which is guided by human-preference-aligned reward from VTON-VLLM. The results are reported in Table 8, and the best results are achieved with $N = 3$.

**VTON-VLLM vs Base VLLM.** To further evaluate the effectiveness of our proposed VTON-VLLM, we compare its capability in VTON assessment with that of the base VLLM (i.e., Pixtral-12B [1]) without fine-tuning on human feedback data. We sample 100 challenging garment-person pairs from VITON-HD, where OOTDiffusion [46] struggles to produce satisfactory outputs, and generates 10 candidate images for each pair using different random seeds. Human annotators then compare the highest-rated images selected by both models from the 10 candidates to determine which is better. As a result, our VTON-VLLM achieves a win rate of 74.17% compared to the base VLLM. Some examples are shown in Figure 7. It is evident that our VTON-VLLM consistently selects the

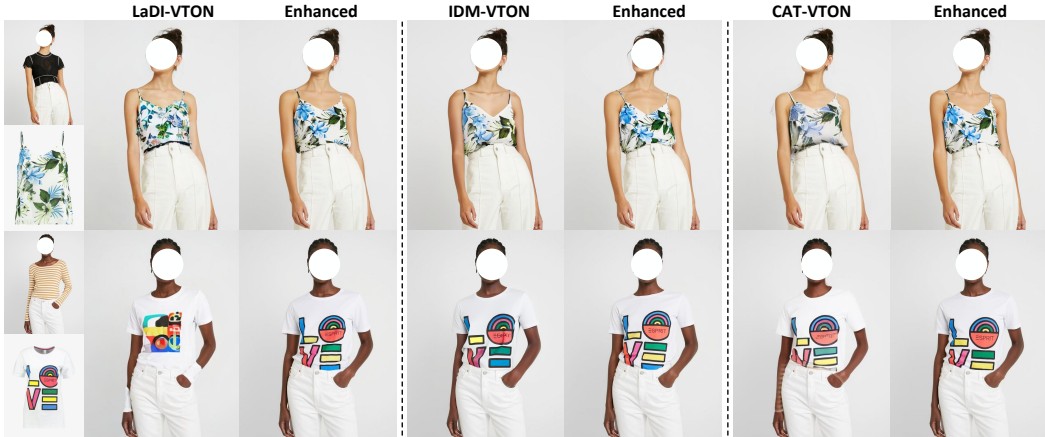

Figure 4: Qualitative results on VTON-HD dataset in unpaired settings. For each pair, the left image is generated by a baseline model (i.e., LaDI-VTON, IDM-VTON, CAT-VTON), and the right image is enhanced by incorporating our proposed VRM-Instruct and TTS module powered by VTON-VLLM.

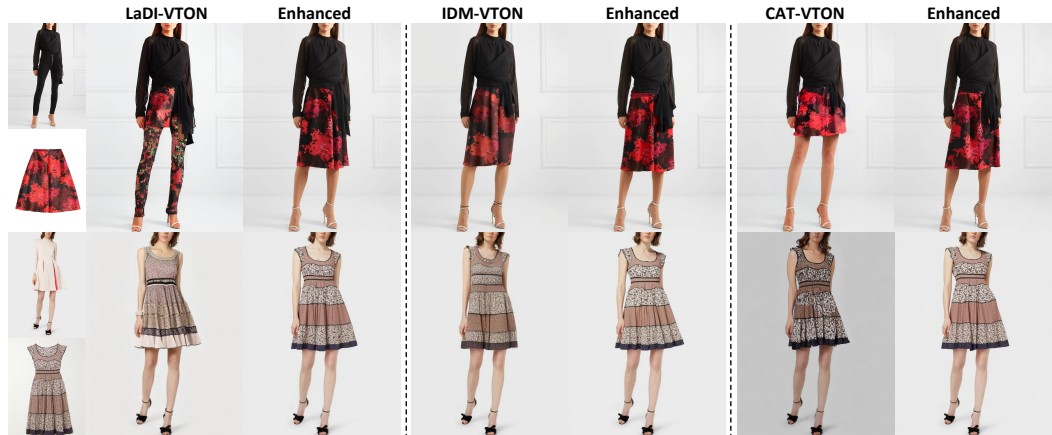

Figure 5: Qualitative results on DressCode dataset in unpaired settings. For each pair, the left image is generated by a baseline model (i.e., LaDI-VTON, IDM-VTON, CAT-VTON), and the right image is enhanced by incorporating our proposed VRM-Instruct and TTS module powered by VTON-VLLM.

perceptually superior image, whereas the base VLLM often overlooks subtle garment inconsistency and visual artifacts. This highlights the effectiveness of our fine-tuned VTON-VLLM in aligning VTON evaluations with human judgments. On the other hand, to verify the importance of training VLLMs on human feedback datasets, we conduct experiments on the VITON-HD dataset under two settings: (1) Base-VLLM: VTON refinement model is trained on and guided by the instructions generated by the Base-VLLM without human feedback fine-tuning; (2) VTON-VLLM: VTON refinement model is trained on and guided by the instructions generated by our VTON-VLLM, which is fine-tuned with human feedback. Both models are trained and evaluated on the VITON-HD dataset. The quantitative results are summarized in Table 9. It can be observed that directly leveraging the Base-VLLM to prompt our proposed VTON refinement model can, to some extent, improve the VTON results compared with the base VTON model. Further empolying our VTON-VLLM to align the refinement with human preferences consistently outperform those with the Base-VLLM across all the metrics, which validates the effectiveness of training VTON-VLLM with human feedbacks on VTON images.

**Performances across fine-grained categories.** We reported the performances across different genders, body types and skin tones. Gender, body type, and skin tone are roughly categorized into several classes. The quantitative results on VITON-Bench are summarized in Table 10.

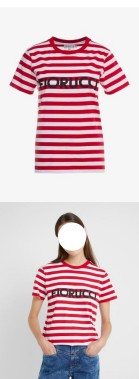

System-Message($\mathbf{X}_0$): You are a professional digital artist. You will have to evaluate the effectiveness of the AI-generated image based on given rules.
Human($\mathbf{X}_q^1$): Please describe characters on upper-body garment in each of two images separately.
Assistant($\mathbf{X}_a^1$): In the first image, the upper-body garment is a horizontally striped red and white T-shirt. The word "FIORUCCI" is printed in bold, black and red letters across the chest area, centered and slightly tilted. In the second image, the T-shirt is the same as in the first image, featuring the horizontally striped red and white pattern. The word "FIORUCCI" is also printed in bold, black and red letters across the chest area, centered and slightly tilted.
Human($\mathbf{X}_q^2$): Now, based on the characters on the upper-body garments in the two images, please determine how well the characters are consistent in the two images, output a score in [0, 5].
Assistant($\mathbf{X}_a^2$): 5.
Human($\mathbf{X}_q^3$): Review your previous responses step-by-step. If you are completely confident that your reasoning and conclusions are correct, respond with "Correct". If you find any errors or are uncertain, respond with "Wrong".
Assistant($\mathbf{X}_a^3$): Correct.

Figure 6: An example of a multi-turn conversational instruction for text character consistency from our human feedback dataset.

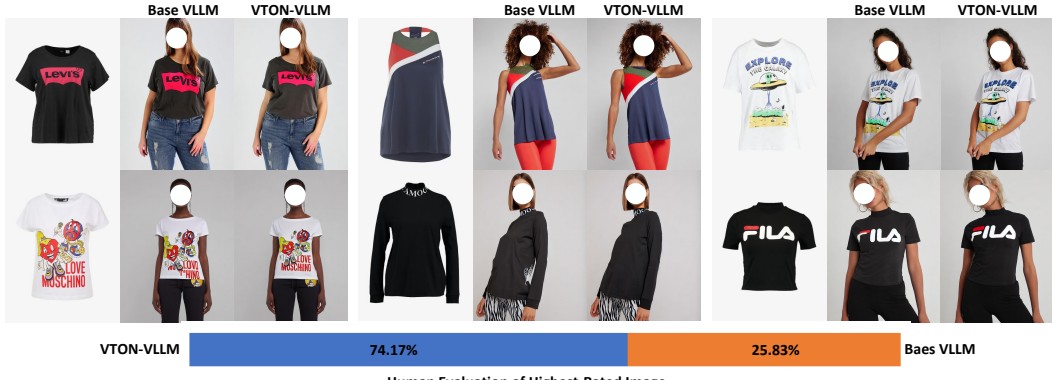

Figure 7: Comparisons between top-ranked images selected by our VTON-VLLM and the base VLLM from 10 candidate synthesized images for each garment-human pair.

### A.3 Dataset Details

**Human Feedback Data.** During the collection of human feedback data, a pre-trained VLLM is first employed to roughly assess garment-person images across various fine-grained dimensions. Human annotators then review and validate these evaluations to ensure accuracy. These annotations are further structured into multi-turn conversational instructions for training our VTON-VLLM. An example instruction for text character consistency is illustrated in Figure 6.

**Synthetic VITON-SYN.** We adopt a similar network to our VRM-Instruct (described in Section 4.1) as the VTON model and train it on VITON-HD to generate VTON data in Section 5.3. Different from VRM-Instruct, the inpainting condition in this VTON model is obtained by combining the garment image $I_g$ and a blank white image $I_\mathbf{1}$, leading to $I_{cond} = I_g \copyright I_\mathbf{1}$. To obtain VITON-SYN, we sample garment images from existing datasets [8, 30, 36] and scene prompts from a pre-defined pool as reference conditions, and synthesize images of persons wearing the corresponding garments

Table 8: Ablation study of iteration count $N$ in our TTS on VITON-HD.

| Model | Garment Consistency | | | | Image Quality | | | |
|---|---|---|---|---|---|---|---|---|
| | Pattern | Characters | Sleeve | Shape | Edge Artifact | Human Pose | SSIM ↑ | LPIPS ↓ |
| 2 | 4.817 | 4.803 | 4.720 | 4.688 | 4.736 | 4.450 | 0.825 | 0.064 |
| **3** | **4.931** | **4.912** | **4.822** | **4.872** | **4.890** | **4.726** | **0.903** | **0.043** |
| 5 | 4.896 | 4.882 | 4.807 | 4.750 | 4.820 | 4.601 | 0.883 | 0.058 |

with the trained VTON model. The derived garment-person pairs are further rigorously filtered by the proposed VTON-VLLM, resulting in a high-quality and human-preference-aligned training dataset. Figure 8 illustrates the data construction pipeline. Overall, VITON-SYN consists of 100,000 garment-person pairs spanning ten different scenarios, which is significantly larger and more diverse than VITON-HD dataset (13,679 indoor samples). Figure 10 showcases various prompts along with their corresponding synthesized images from VITON-SYN.

**New Test Set VITON-Bench.** Unlike existing benchmarks such as VITON-HD and DressCode, our VITON-Bench emphasizes more on challenging VTON scenarios. These include uniquely styled clothing, complex human poses (e.g., sitting or lying down), and diverse, intricate outdoor environments. Figure 9 illustrates some examples from our VITON-Bench, highlighting its greater scene diversity and increased difficulty compared to previous benchmarks (see Figure 4 and 5).

Table 9: Quantitative comparisons between Base-VLLM and VTON-VLLM on VITON-HD dataset.

| Train/Test | VITON-HD/VITON-HD | | | | Average Preference-Aware Scores | | | |
|---|---|---|---|---|---|---|---|---|
| Method | SSIM ↑ | LPIPS ↓ | FID ↓ | KID ↓ | $GC_p$ ↑ | $IQ_p$ ↑ | $GC_u$ ↑ | $IQ_u$ ↑ |
| **Base-VTON** [10] | 0.870 | 0.057 | 9.015 | 0.670 | 4.713 | 4.765 | 4.510 | 4.420 |
| w/ **Base-VLLM** | 0.880 | 0.054 | 8.976 | 0.660 | 4.843 | 4.880 | 4.698 | 4.625 |
| w/ **VTON-VLLM** | **0.891** | **0.052** | **8.923** | **0.652** | **4.895** | **4.900** | **4.733** | **4.738** |

Table 10: Quantitative comparisons across fine-grained categories on VITON-Bench. VRM-Instruct and TTS are short for VTON refinement model with fine-grained instructions and test-time scaling.

| Categories | Gender | | Skin Tone | | | | Body Type | | |
|---|---|---|---|---|---|---|---|---|---|
| Method | Male SSIM ↑ | Female SSIM ↑ | Black SSIM ↑ | White SSIM ↑ | Yellow SSIM ↑ | Brown SSIM ↑ | Slim SSIM ↑ | Average SSIM ↑ | Overweight SSIM ↑ |
| **Base-VTON** [10] | 0.7500 | 0.7514 | 0.7562 | 0.7538 | 0.7565 | 0.7241 | 0.7623 | 0.7211 | 0.7603 |
| **Base + VRM-Instruct + TTS** | **0.7763** | **0.7963** | **0.7700** | **0.7815** | **0.7570** | **0.8206** | **0.7970** | **0.7521** | **0.7830** |

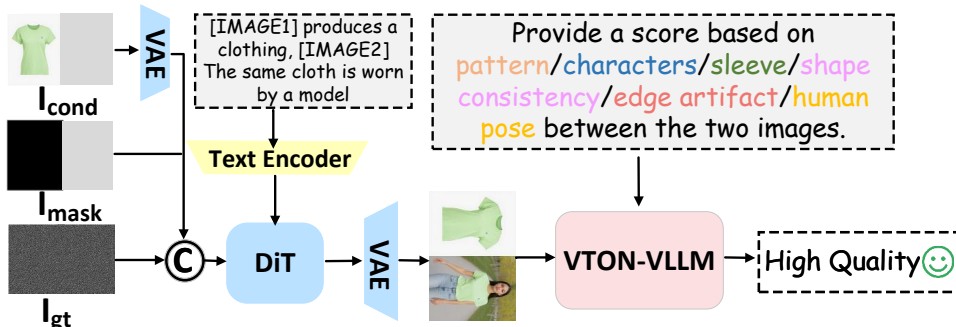

Figure 8: Data construction pipeline of synthetic VITON-SYN. The instruction provided to VTON-VLLM is simplified here.

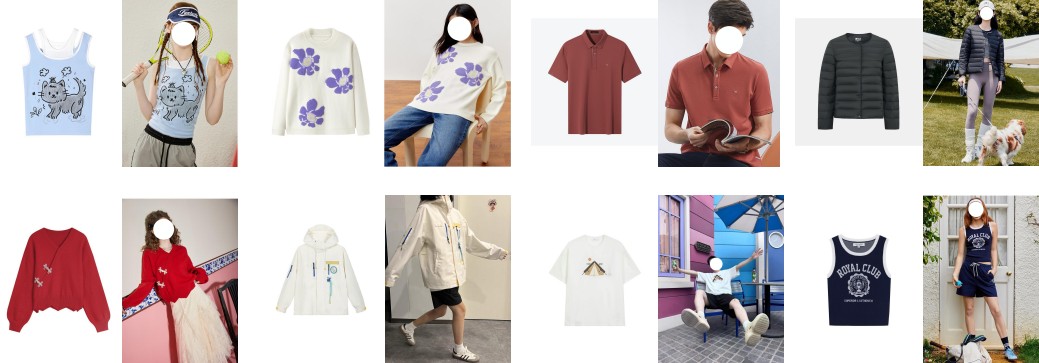

Figure 9: Examples from our VITON-Bench.

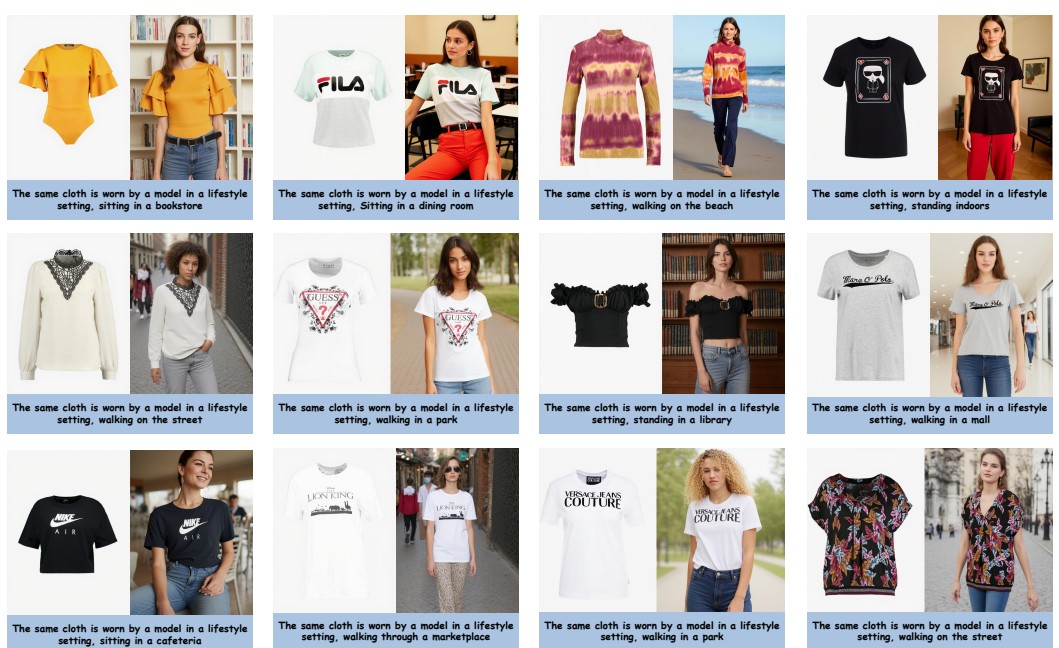

Figure 10: Different prompts along with their corresponding synthesized images from VITON-SYN.

