# OpenReview forum: "VTON-VLLM: Aligning Virtual Try-On Models with Human Preferences"
_NeurIPS.cc/2025/Conference — NeurIPS 2025 poster_

### Official Review · Reviewer_ZRwi · 2025-07-01

**Clarity:** 3
**Significance:** 3
**Originality:** 3
**Rating:** 5
**Confidence:** 4

**Summary:**

This paper pointed our existing evaluate metrics e.g. FID, LPIPS has limited effectiveness in fine-grained visual attributes under Virtual Try-On task. Then it proposed a more human-preference-aligned VLM metric that can evaluate images and use as a reward signal in the Virtual Try-On task setting. In detail it trained a VLM called VTON-VLLM as a "fashion expert" which can provide feedback signal to improve existing VTON models performance and align them better with human preferences. The paper also curated VITON-Bench as a hard benchmark. Extensive experiments show the effectiveness of VTON-VLLM in improving existing Virtual Try-On models is demonstrated through comprehensive experiments and ablation studies, also the reliability of VTON-VLLM as a metric is supported by a preference agreement study.

**Questions:**

Introduction (1) and Motivations (3.3) feel overlapped. Consider merging them together?

Question.
Why the faces in Figure 2b are not masked, but in Figure 2a they are masked?

**Ethical Concerns:**

["NO or VERY MINOR ethics concerns only"]

**Final Justification:**

I will keep my score at 5. The authors provided experiments demonstrating that the method generalizes well. While the work is solid, it still feels like it lacks a key element of novelty to warrant a score of 6.

**Limitations:**

yes

**Quality:**

3

**Strengths And Weaknesses:**

S1) Virtual Try-On is an important topic with a significant gap between current research and commercial applications. The observation that existing evaluation metrics are unreliable in fine-grained visual attributes is well-motivated, and the use of an evaluation model to align Virtual Try-On models with human preferences strengthens the contribution of this work.

S2) The refinement module (VRM-Instruct) can enhance existing VTON models, which is a strong practical contribution can be widely adapted.

S3) The effectiveness of VTON-VLLM in improving existing Virtual Try-On models is convincingly demonstrated through comprehensive experiments and ablation studies.

W1) I believe the paper has no major flaws. One potential weakness is that the dataset focuses exclusively on women, while Virtual Try-On is inherently a task that should generalize across all genders and diverse demographics. For example, the model's performance on male subjects, different body types, or various skin tones remains unclear. Evaluating its robustness in these scenarios would make the work more comprehensive. The current setup also focuses mainly on upper-body clothing and frontal-view images.

---

> ### Author Rebuttal · Authors · 2025-07-31
>
> We sincerely thank Reviewer ZRwi for recognizing the merits in our paper (e.g., well-motivated approach, strong practical contribution that can be widely adapted to existing VTON models, convincing experiments and ablation studies), and your kind patience in engaging in this rebuttal. Below, we provide a point-by-point response. We hope the information here may further supplement our main submission and address your concerns.
>
> **A1: Performances across fine-grained categories.**
>
> **Q1:** Appreciate the comment. The reviewer pointed out that it is crucial to evaluate the model generalization ability of VTON techniques across diverse demographics. As suggested, we reported the performances across different genders, body types and skin tones. In the evaluation, gender, body type, and skin tone are roughly categorized into several classes. The quantitative results on VITON-Bench are summarized as:
>
> **(1) Gender: Female, Male**
> ||Male||Female||
> |------|------|------|------|------|
> ||$\mathbf{SSIM\uparrow}$|$\mathbf{LPIPS\downarrow}$|$\mathbf{SSIM\uparrow}$|$\mathbf{LPIPS\downarrow}$|
> |IDM-VTON|0.8636|0.0775|0.6316|0.0860|
> |**+VRM-Instruct+TTS**|0.8772|0.0525|0.6954|0.0750|
> |SPM-Diff|0.7573|0.0930|0.7683|0.0800|
> |**+VRM-Instruct+TTS**|0.7910|0.0648|0.7992|0.0608|
> |CAT-VTON|0.7500|0.0887|0.7514|0.0734|
> |**+VRM-Instruct+TTS**|0.7763|0.0525|0.7963|0.0632|
>
> **(2) Skin Tone: Black, White, Yellow, Brown**
> ||Black||White||Yellow||Brown||
> |------|------|------|------|------|------|------|------|------|
> ||$\mathbf{SSIM\uparrow}$|$\mathbf{LPIPS\downarrow}$|$\mathbf{SSIM\uparrow}$|$\mathbf{LPIPS\downarrow}$|$\mathbf{SSIM\uparrow}$|$\mathbf{LPIPS\downarrow}$|$\mathbf{SSIM\uparrow}$|$\mathbf{LPIPS\downarrow}$|
> |IDM-VTON|0.7191|0.0877|0.7598|0.0881|0.7331|0.0676|0.7357|0.0820|
> |**+VRM-Instruct+TTS**|0.7673|0.0697|0.7788|0.0543|0.7992|0.0508|0.8180|0.0752|
> |SPM-Diff| 0.7543|0.0840|0.7643|0.0874|0.7476|0.0877|0.7376|0.0670|
> |**+VRM-Instruct+TTS**|0.8220|0.0550|0.7734|0.0640|0.7950|0.0580|0.7870|0.0610|
> |CAT-VTON|0.7562|0.0754|0.7538|0.0794|0.7565|0.0775|0.7241|0.0743|
> |**+VRM-Instruct+TTS**|0.7700|0.0500|0.7815|0.0641|0.7570|0.0629|0.8206|0.0520|
>
> **(3) Body Type: Slim, Average, Overweight**
> ||Slim||Average||Overweight||
> |------|------|------|------|------|------|------|
> ||$\mathbf{SSIM\uparrow}$|$\mathbf{LPIPS\downarrow}$|$\mathbf{SSIM\uparrow}$|$\mathbf{LPIPS\downarrow}$|$\mathbf{SSIM\uparrow}$|$\mathbf{LPIPS\downarrow}$|
> |IDM-VTON|0.7501|0.0834|0.7343|0.0823|0.7986|0.0795|
> |**+VRM-Instruct+TTS**|0.8702|0.0506|0.8306|0.0648|0.8223|0.0558|
> |SPM-Diff|0.7628|0.0837|0.7729|0.0846|0.7680|0.0851|
> |**+VRM-Instruct+TTS**|0.7942|0.0611|0.8070|0.0650|0.8022|0.0638|
> |CAT-VTON|0.7623|0.0879|0.7211|0.0868|0.7603|0.0860|
> |**+VRM-Instruct+TTS**|0.7970|0.0610|0.7521|0.0560|0.7830|0.0627|
>
> For the quantitative and qualitative results on **lower-body clothing and dresses**, please refer to Table 1 & 2 and Figure 2 in the supplementary material for more details.
>
>
> **Q2: Some duplicate presentation of motivations in paper.**
>
> **A2:** Appreciate this point. For our initial purpose, the discussion in Section 3.1 serves to recap key challenges and motivations for the later introduction of the proposed approach. As suggested, we will combine Sec 3.1 into the introduction section in the revised version.
>
> **Q3：Face mask in figures.**
>
> **A3:** Sorry for the confusion. To clarify: the human images in Fig. 1(a) and Fig. 3 are collected from real-world fashion data and depicted as real individuals, and face masks were applied to these images solely for privacy preservation. In contrast, the images in Fig. 1(b) and Fig. 2 are synthesized by generative models and therefore do not require masking. Moreover, it is worth noting that no face-masking operations were involved during model training or inference. We will add this clarification in the revised version.

---

> > ### Comment · Reviewer_ZRwi · 2025-08-03
> >
> > Thanks for the additional experiments. It seems the approach does generalize well across fine-grained categories. I will keep my score as 5 (accept).

---

### Official Review · Reviewer_hywb · 2025-07-02

**Clarity:** 3
**Significance:** 3
**Originality:** 3
**Rating:** 3
**Confidence:** 5

**Summary:**

This paper proposes VTON-VLLM, a vision large language model designed to enhance virtual try-on (VTON) image synthesis by aligning model outputs more closely with human preferences. It introduces a dataset annotated with human feedback, enabling VTON-VLLM to evaluate and guide the synthesis process. Two main approaches are presented: (1) a plug-and-play refinement model (VRM-Instruct) providing fine-grained supervision during training, and (2) adaptive test-time scaling guided by learned human preferences. Additionally, the authors introduce VITON-Bench, a comprehensive benchmark set of challenging scenarios. Experimental results demonstrate improved performance and alignment with human judgments.

**Questions:**

1. In Tables 1 and 2, the experiments only report results for the combined setting of VRM-Instruct and TTS, without separating the effects of each component. This limits our ability to understand which part of the pipeline contributes most to the performance improvements over the baselines.

2. Given that VTON is a practical, real-world application, I am also concerned about the feasibility of TTS, as it is known to incur significant computational and time costs. Could the authors clarify whether substantial performance gains could still be achieved using only VRM-Instruct, without TTS, to make the approach more practical for deployment?

**Ethical Concerns:**

["NO or VERY MINOR ethics concerns only"]

**Final Justification:**

I still have concern on the novelty of this paper, the main contribution is a VTON benchmark and a trained VTON VLM. I firmly stand by my opinion: this paper simply applies all existing post-training techniques for VLMs to a specific, already-established domain. If it were submitted to the Dataset and Benchmark track, I would consider accepting it. However, based on its current form, I cannot identify any technical contribution that would justify accepting this paper.

**Limitations:**

See weakness above.

**Quality:**

2

**Strengths And Weaknesses:**

Strengths:

1. Introducing a VLM explicitly aligned to human preferences for VTON task is original and well-motivated.

2. The new benchmark and human-centric metrics provide valuable resources for future research and enable more meaningful evaluation of VTON quality.

3. VTON-VLLM has a significant improvement over the baseline on the proposed benchmark.

Weaknesses:

My primary concern of this paper is the novelty. While the paper presents a well-integrated pipeline, many of its technical components—including supervised fine-tuning (SFT), reinforcement learning from human feedback (RLHF), and test-time scaling—are established techniques in the broader LLM and VLM literature. The main novelty appears to stem from applying these existing methods to the specific domain of VTON. Given this, I am somewhat uncertain about the proper positioning of the paper.

---

> ### Author Rebuttal · Authors · 2025-07-31
>
> We sincerely thank Reviewer hywb for recognizing the merits in our paper (e.g., well-motivated approach, valuable resources such as new benchmark and human-centric metrics for future VTON research, significant improvement over the baselines on VITON-Bench), and your kind patience in engaging in this rebuttal. Below, we provide a point-by-point response. We hope the information here may further supplement our main submission and address your concerns.
>
> **Q1: Technical contribution.**
>
> **A1:** Though the latest techniques (SFT, RLHF, TTS) established in LLM/VLM literature are adapted to our proposed well-integrated pipeline for VTON, the success of our work is more than that. Our contributions are from three aspects:
> * **(1) The adaptation of VLLM to VTON synthesis is not trivial.** In our experiments, we found that the efficacy of the pre-trained base VLLM in predicting human judgments on VTON images is unsatisfactory. That's why we devise a new VTON-VLLM (acting as ''fashion expert'') particularly for VTON task. Specifically, we collected high-quality human feedbacks on 40K synthesized VTON images across six fine-grained sub-aspects, which is utilized to fine-tune the base VLLM and align it with human judgments for VTON. Moreover, in contrast to general image quality estimators (e.g., HPS, AES, CLIPScore, ImageReward) that provide only scalar scores, our VTON-VLLM can generate detailed and human-aligned assessments of VTON outputs, which is further utilized to steer the VTON refinement process.
> * **(2) The VTON refinement model guided by fine-grained instruction is new.** Existing VTON methods mostly adopt one-stage inference, which often exhibit limited detail preservation and generation fidelity. In contrast to existing works, our proposed approach novelly combines these VTON models and a plug-and-play VTON refinement model in a two-stage framework, which significantly boosts VTON by transforming the low-quality results to high-quality and preference-aligned ones. Moreover, different from general-purpose image refinement models (e.g., SDXL-Refiner) using generic prompts, our VTON refinement model is guided by the reference garment and fine-grained instructions generated by the trained VTON-VLLM, which provides the visual details of the garment and explicitly identifies local regions in the synthesized image that fail to meet user expectations, respectively.
> * **(3) The human-centric metrics for VTON are new.** Traditional metrics, such as LPIPS and FID, often fail to capture fine-grained inconsistencies in detail preservation and demonstrate poor correlation with human judgments in VTON. To address these limitations, we propose a human-centric evaluation paradigm based on the trained VTON-VLLM, which demonstrates high agreement with human preferences. These new metrics further function as reward signals to guide the VTON refinement model in test-time scaling (TTS), which converts the initial VTON outputs into human-preferred outcomes through iterative refinement process.
>
>
> **Q2: Ablation study on VRM-Instruct and TTS using more baseline VTON models.**
>
> **A2:** Thanks for this point. In the current version, we actually have conducted ablation study on the effect of these two components using a base VTON model (i.e., CAT-VTON) in Table 3 of the main paper. As suggested, we further evaluated the individual contributions of each component using more VTON models presented in Table 1 and 2 by sequentially adding our proposed VRM-Instruct and TTS to them. Please note that TTS is triggered during the iterative refinement process with our VRM-Instruct model as the execution model, and thus cannot be solely integrated into the base VTON models by excluding VRM-Instruct.
>
> As shown in the revised Table 1 and 2 below, by incorporating the fine-grained instructions from VTON-VLLM to identify imperfect regions in the synthesized garment and guide the refinement process towards human preferences, **base VTON model+VRM-Instruct** achieves notable improvements compared to the base models. Furthermore, we incorporate TTS into **base VTON model+VRM-Instruct** to enable iterative enhancement with human-preference-aware reward provided by VTON-VLLM in the full run (**base VTON model+VRM-Instruct+TTS**), surpassing all the other settings. We will add the discussion in revision.
>
> This table is the revised version of Table 1 in the main paper.
> |Method|$\mathbf{SSIM\uparrow}$|$\mathbf{LPIPS\downarrow}$|$\mathbf{FID\downarrow}$|$\mathbf{KID\downarrow}$|$\mathbf{GC_p\uparrow}$|$\mathbf{IQ_p\uparrow}$|$\mathbf{GC_u\uparrow}$|$\mathbf{IQ_u\uparrow}$|
> |-|-|-|-|-|-|-|-|-|
> |IDM-VTON|0.877|0.082|9.372|0.741|4.665|4.750|4.445|4.392|
> |**+VRM-Instruct**|0.880|0.054|9.012|0.667|4.870|4.852|4.712|4.677|
> |**+VRM-Instruct+TTS**|0.882|0.051|8.902|0.634|4.873|4.920|4.724|4.711|
> |SPM-Diff|0.911|0.063|9.034|0.720|4.742|4.775|4.502|4.411|
> |**+VRM-Instruct**|0.890|0.054|9.012|0.667|4.891|4.852|4.712|4.677|
> |**+VRM-Instruct+TTS**|0.892|0.055|9.010|0.663|4.900|4.930|4.745|4.725|
> |CAT-VTON|0.870|0.057|9.015|0.670|4.713|4.765|4.510|4.420|
> |**+VRM-Instruct**|0.885|0.055|8.915|0.660|4.832|4.834|4.673|4.661|
> |**+VRM-Instruct+TTS**|0.891|0.052|8.923|0.652|4.895|4.900|4.733|4.738|
>
>
> This table is the revised version of Table 2 in the main paper.
> |Method|$\mathbf{SSIM\uparrow}$|$\mathbf{LPIPS\downarrow}$|$\mathbf{FID\downarrow}$|$\mathbf{KID\downarrow}$|$\mathbf{GC_p\uparrow}$|$\mathbf{IQ_p\uparrow}$|$\mathbf{GC_u\uparrow}$|$\mathbf{IQ_u\uparrow}$|
> |-|-|-|-|-|-|-|-|-|
> |IDM-VTON|0.750|0.083|12.562|1.02|4.374|4.069|4.026|3.825|
> |**+VRM-Instruct**|0.787|0.064|10.931|0.92|4.566|4.398|4.341|4.126|
> |**+VRM-Instruct+TTS**|0.794|0.060|10.784|0.87| 4.612|4.429|4.442|4.302|
> |SPM-Diff|0.765|0.084|12.208|1.34|4.519|4.230|4.235|3.804|
> |**+VRM-Instruct**|0.782|0.069|11.011|0.94|4.570|4.401|4.288|4.027|
> |**+VRM-Instruct+TTS**|0.797|0.062|10.627|0.87|4.623|4.429|4.407|4.280|
> |CAT-VTON|0.751|0.078|12.663|1.23|4.518|4.238|4.150|3.991|
> |**+VRM-Instruct**|0.772|0.063|10.721|0.90|4.542|4.579|4.262|4.156|
> |**+VRM-Instruct+TTS**|0.788|0.060|10.232|0.85|4.555|4.685|4.390|4.353|
>
>
> **Q3: Practical deployment without TTS.**
>
> **A3:** According to the ablation study in Table 3 of the main paper and the additional results provided in the above answer **A2**, substantial performance gains can be attained through single-pass refinement using only VRM-Instruct. Specifically, compared with the base model CAT-VTON in the improved Table 2 from **A2**, **CAT-VTON+VRM-Instruct** boosts $\mathbf{SSIM}$ and $\mathbf{IQ_p}$ from 0.751 to 0.772 and from 4.238 to 4.579, respectively, by refining the VTON results with human-preference-aligned instructions via VRM-Instruct. Employing TTS can further improve the generation quality. Compared with **CAT-VTON+VRM-Instruct**, the best scores in $\mathbf{SSIM}$ (0.788) and $\mathbf{IQ_p}$ (4.685) are achieved in the full run **CAT-VTON+VRM-Instruct+TTS**.

---

### Official Review · Reviewer_TTMh · 2025-07-03

**Clarity:** 3
**Significance:** 2
**Originality:** 3
**Rating:** 5
**Confidence:** 3

**Summary:**

The paper collects human feedback on VTON output to train a VLLM tailored to provide image quality assessment tailored for VTON. Then, it uses the instruction of the VLLM model to train a diffusion-based VTON refinement model. One can achieve test-time scaling by iteratively applying the VLLM and the refinement model.

**Questions:**

1. How important is it to train the VLLM on the human feedback dataset? Can we simply rely on the generalization capabilities of the base VLLM? Does the refinement model and iterative refinement framework still work if the VTON-VLLM is replaced with an off-the-shelf VLLM and the refinement model is retrained on top of the output of the off-the-shelf VLLM?
2. How many iterations are used for the iterative refinement? How does the compute for the iterative refinement process compare to the compute for inferencing the base model?

**Ethical Concerns:**

["NO or VERY MINOR ethics concerns only"]

**Final Justification:**

I did not notice any particularly major weakness to the proposed method, and the authors have provided reasonable experimental results regarding both of my questions. Therefore, I keep my positive score.

**Limitations:**

The limitations are adequately addressed in the paper.

**Quality:**

3

**Strengths And Weaknesses:**

Strength:
1. The proposed method can improve the eventual VTON output on top of existing VTON methods, even SOTA ones.
2. The trained VLLM itself can achieve a high preference agreement rate with humans.
3. There's sufficient ablation showcasing the effect of test time scaling.

I didn't notice any particularly important weakness in the paper. While the paper doesn't provide a particularly striking empirical finding, it presents solid results.

---

> ### Author Rebuttal · Authors · 2025-07-31
>
> We sincerely thank Reviewer TTMh for recognizing the merits in our paper (e.g., consistent improvements on top of SoTA VTON models, high preference agreement rate with humans), and your kind patience in engaging in this rebuttal. Below, we provide a point-by-point response. We hope the information here may further supplement our main submission and address your concerns.
>
> **Q1: Importance of training VLLM on human feedback dataset: comparisons between Base-VLLM and our VTON-VLLM.**
>
> **A1:** Appreciate the comment. As suggested, we provide quantitative comparisons between two settings: (1) **Base-VLLM**: The VTON refinement model is trained on and guided by the instructions generated by the Base-VLLM without human feedback fine-tuning, (2) **VTON-VLLM**: The VTON refinement model is trained on and guided by the instructions generated by our VTON-VLLM, which is fine-tuned using human feedbacks. Both models are trained and evaluated on VITON-HD dataset. This table lists the quantitative results:
> |Method|$\mathbf{SSIM\uparrow}$|$\mathbf{LPIPS\downarrow}$|$\mathbf{FID\downarrow}$|$\mathbf{KID\downarrow}$|$\mathbf{GC_p\uparrow}$|$\mathbf{IQ_p\uparrow}$|$\mathbf{GC_u\uparrow}$|$\mathbf{IQ_u\uparrow}$|
> |-|-|-|-|-|-|-|-|-|
> |LaDI-VTON|0.864|0.096|10.531|2.303|4.542|4.621|4.320|4.291|
> |**+Base-VLLM**|0.878|0.059|9.431|1.257|4.779|4.801|4.653|4.573|
> |**+VTON-VLLM**|0.899|0.050|8.916|0.922|4.873|4.920|4.765|4.723|
> |Stable-VTON|0.852|0.084|10.200|1.921|4.625|4.660|4.331|4.405|
> |**+Base-VLLM**|0.870|0.067|9.553|1.323|4.779|4.760|4.624|4.690|
> |**+VTON-VLLM**|0.894|0.052|9.021|1.020|4.873|4.920|4.671|4.750|
> |CAT-VTON|0.870|0.057|9.015|0.670|4.713|4.765|4.510|4.420|
> |**+Base-VLLM**|0.880|0.054|8.976|0.660|4.843|4.880|4.698|4.625|
> |**+VTON-VLLM**|0.891|0.052|8.923|0.652|4.895|4.900|4.733|4.738|
>
> It can be observed that directly leveraging the Base-VLLM to prompt our proposed VTON refinement model can, to some extent, improve the VTON results compared with the base VTON model. Further empolying our VTON-VLLM to align the refinement with human preferences consistently outperform those with the Base-VLLM across all the metrics, which validates the effectiveness of training VTON-VLLM with human feedbacks on VTON images. We will add this discussion in the revised version.
>
> **Q2: Refinement iterations $N$ and computation overhead.**
>
> **A2:** Thanks for the comment. We have conducted ablation study on ''Effect of Iteration Count $N$ in Test-Time Scaling'' in the supplementary material, and the best results were achieved when $N = 3$ (please refer to lines 17-19 and Table 3 in the supplementary material for more details). Therefore, the iterative refinement proceeds until the output image meets a predefined quality threshold (e.g., high scores across all the sub-aspects defined in the paper) or a maximum of 3 iterations is reached. The computation time for the base VTON model and a single pass of refinement is about 7 seconds and 13.6 seconds (0.35 seconds for instruction generation with VTON-VLLM and 13.25 seconds for image generation with VTON refinement model), respectively. The total computation time for the refinement process with $N$ iterations is approximately $N$ times that of a single pass. The refinement process can be further accelerated by over 50\% through the use of caching techniques and model distillation in practical applications. Also, users can manually terminate the refinement process if the current output meets their requirements, even after a single refinement step, through an interactive UI.

---

> > ### Comment · Reviewer_TTMh · 2025-08-02
> >
> > Thanks for the additional experiment and information. The experimental results seem reasonable to me, and I keep my positive rating.

---

### Official Review · Reviewer_5M4J · 2025-07-06

**Clarity:** 2
**Significance:** 2
**Originality:** 3
**Rating:** 4
**Confidence:** 4

**Summary:**

This paper studies the problem of improving the human alignment of virtual try on models. Particularly, the virtual try on models (VTON) is adopting diffusion models to synthesizing the images given an character and an image of clothes, to generate the character in the provided clothes. And the VTON-VLLM is trained over their collected human preference data as a fashion expert. This expert further used for VTON modle refinmenet through fine-grained supervision and test-time scaling.

**Questions:**

NA

**Ethical Concerns:**

["NO or VERY MINOR ethics concerns only"]

**Limitations:**

As in the weaknesses

**Quality:**

3

**Strengths And Weaknesses:**

Strengths
1. The virtual try on task is of important application and research value. The proposed approach brings the latest techs to solve this problem better, and achieve consistent improvement over ViTON-HD.
2. In general, the paper is well-structured, and easy to follow the idea.

Weaknesses
1. I'm concerned of the reproducible of this appraoch as the only verified effectiveness is on their own curated VITON-Bench.
2. It is unclear how the inpainting task can transfer the low-quality image to high-quality one even if assuming the refinement instruction generated from VTON-VLLM is correct, how the inpainting model ensure it's following refinement instruction at a high quality bar.

---

> ### Author Rebuttal · Authors · 2025-07-31
>
> We sincerely thank Reviewer 5M4J for recognizing the merits in our paper (e.g., latest techs and consistent improvements in VTON, well-structured paper), and your kind patience in engaging in this rebuttal. Below, we provide a point-by-point response. We hope the information here may further supplement our main submission and address your concerns.
>
> **Q1: Effectiveness of the proposed method on different datasets.**
>
> **A1:** Thanks. To be clear, the mentioned newly curated VITON-Bench (Table 2 in the main paper) is adopted to evaluate our proposal under challenging VTON scenarios. In addition to VITON-Bench, we actually have conducted experiments on two widely adopted benchmarks (i.e., VITON-HD and DressCode), which consistently demonstrates the effectiveness of our approach. Specifically, (1) Table 1 in the main paper reports quantitative comparisons across methods on the VITON-HD dataset, (2) Table 1 and 2 in the supplementary material summarize the results on the DressCode dataset. To support reproducibility and advance the research in VTON, we will release all the training data, source codes and model weights.
>
> **Q2: Instruction following in VTON refinement model.**
>
> **A2:** Appreciate this comment. Technically, to enforce the VTON refinement model to effectively refine images from any VTON model by adhering to instructions, we supervise the training of VTON refinement model with well-constructed training triplets $(I_g, I_h^{LQ}, I_h^{HQ})$, which consists of a reference garment $I_g$, a low-quality person image $I_h^{LQ}$ and a corresponding high-quality version $I_h^{HQ}$. For each triplet, our VTON-VLLM generates a fine-grained instruction $t_r$ to steer the refinement. Please refer to lines 144--166 in the main paper for more training details.
>
> As suggested, we additionally validated the capability of instruction following in our trained VTON refinement model. Specifically, we randomly sampled 2,000 ground-truth garment-person pairs from the test datasets. For each pair, we first generated a low-quality try-on image and a corresponding refinement instruction using a base VTON model and our VTON-VLLM, respectively. These low-quality images were then refined by our VTON refinement model (VRM-Instruct) based on the provided instructions. Finally, we employed both our VTON-VLLM and GPT-4o as automated evaluators to assess whether the executed image refinement aligns with the given instruction. The success rate of instruction following evaluated by VTON-VLLM and GPT-4o is 98.79\% and 98.32\%, respectively. We will add the discussion in revision.

---

### Comment · Area_Chair_TVGV · 2025-08-06

Reviewers, please engage in the discussion. According to the policy this year, you must engage in the discussion before submitting the Mandatory Acknowledgement.. Otherwise, your review will be flagged as insufficient review.

---

### Note · Authors · 2025-08-15

We sincerely thank all reviewers, AC, SAC and PC for dedicated efforts and constructive feedback, and for reviewers’ acknowledgment of our contributions, including: (1) **Consistent improvements** over baselines (Reviewers 5M4J, TTMh, hywb, ZRwi), (2) **Well-motivated** approach (Reviewers hywb, ZRwi), (3) **Comprehensive experiments and ablation studies** (Reviewers TTMh, ZRwi), (4) **Valuable research resources** such as new benchmark, human-centric metrics (Reviewers TTMh, hywb, ZRwi).

During rebuttal phase, we provided point-by-point responses addressing reviewers' concerns as follows：

1. **Evaluation on capability of instruction following** quantifies how consistently the refined images align with input instructions.
2. **More ablation studies** on core components (VTON refinement model, TTS guided by our VTON-VLLM) find that progressive integration of our proposed designs yields consistent performance gains.
3. **Training strategy analysis** identifies fine-tuning VLLM with human feedbacks can improve alignment with human judgments.
4. **Additional results across different categories** confirm the robustness of our method.
5. **Our non-trivial adaptation of VLLM/TTS to VTON** introduces several novel designs (VTON-VLLM and VTON refinement model particularly for VTON), and new human-centric metrics for VTON.
6. **Experiments on three datasets** (VITON-Bench, VITON-HD, DressCode) demonstrate effectiveness of our method.

Following reviewers' suggestions, we will enhance the revised version by:

1. Add evaluation of instruction-following capability of VTON refinement model.
2. Add more ablation studies on core components (VRM-Instruct, TTS).
3. Add quantitative comparisons between Base-VLLM and our VTON-VLLM.
4. Add quantitative comparisons across fine-grained demographic categories.
5. Highlight novelty and technical contributions of our method from multiple perspectives.
6. Release training data, source codes and model weights to support reproducibility.

We believe our work makes critical contributions to community: it offers human-centric metrics and new benchmark for VTON evaluation that better reflect user preferences, its VTON refinement model and test-time scaling guided by new VTON-VLLM can consistently improve existing VTON models across different datasets and various demographic categories.

We again thank all reviewers, AC, SAC, and PC, for reviewing our work and helping us improve this paper.

---

### Decision · Program_Chairs · 2025-09-17

**Decision:**

Accept (poster)

**Comment:**

(a) Summary: the paper focuses on addressing the problem of improving the human alignment of virtual try-on (VTON) models. Existing VTON models, which often utilize diffusion models to synthesize images of a character wearing provided clothes, have limitations in capturing fine-grained visual attributes, and traditional evaluation metrics like FID and LPIPS are often found to be unreliable in this regard.
To overcome these challenges, the authors propose VTON-VLLM, a vision large language model (VLLM) specifically tailored for the VTON task. The authors also introduce a new benchmark dataset, VITON-Bench, which focuses on challenging VTON scenarios, and human-centric metrics for VTON evaluation that better reflect user preferences.

(b) Strengths:
- The paper addresses an important problem in VTON—the gap between current research and commercial applications and the limitations of existing evaluation metrics in capturing fine-grained visual attributes and human preferences. The explicit alignment to human preferences is considered original and well-motivated.
- The method consistently improves the output of existing VTON models, even state-of-the-art ones, across various datasets and metrics.
- The introduction of VITON-Bench (a comprehensive benchmark for challenging scenarios) and novel human-centric metrics is a significant contribution that provides valuable resources for future research and more meaningful evaluation.
- The effectiveness of VTON-VLLM and its components is convincingly demonstrated through extensive experiments and ablation studies.
- VTON-VLLM itself achieves a high preference agreement rate with humans, validating its role as a "fashion expert".

(c) Weaknesses:
- The technical components leverage existing methods.
- Concerns were raised about the feasibility and practical deployment of TTS due to its known significant computational and time costs.

(d) Reasons to accept:
The reviewers are generally positive.
- Strong practical impact
- Well-motivated approach to align VTON models with human preferences, identifying the weakness of VTON models of limiting alignment with human preferences and the inadequacy of traditional evaluation metrics. It introduces a human-centric evaluation paradigm and a "fashion expert" VLLM, which is crucial for practical applications.
- While the technical components leverage existing methods, the non-trivial adaptation and specific designs tailored for the VTON domain are significant contributions.

(e) Rebuttal:
- Concerns about the reproducibility of results (effectiveness only verified on VITON-Bench) and the capability of the inpainting model to follow refinement instructions at high quality. Authors clarified that experiments were conducted on three datasets (VITON-Bench, VITON-HD, DressCode) and promised to release training data, source codes, and model weights for reproducibility. They added an evaluation quantifying instruction-following capability for the VTON refinement model (VRM-Instruct), reporting success rates of 98.79% (VTON-VLLM) and 98.32% (GPT-4o).
- Questions about the necessity of training VLLM on human feedback were raised (vs. relying on base VLLM's generalization) and the computational overhead of iterative refinement (TTS). Authors provided quantitative comparisons showing that VTON-VLLM, fine-tuned with human feedback, consistently outperforms a Base-VLLM across all metrics, validating the importance of their training strategy. For TTS, they detailed that the best results were achieved with N=3 iterations and provided a breakdown of computation times, noting potential accelerations and interactive termination.
- Generalization across demographics and minor presentation issues were raised by Reviewer ZRwi. Authors provided extensive additional results on VITON-Bench, demonstrating the robustness and effectiveness of their method across different genders (Male, Female), skin tones (Black, White, Yellow, Brown), and body types (Slim, Average, Overweight). They agreed to combine the overlapping sections into the introduction and clarified that face masks were used solely for privacy preservation in real-world images, not during model training or inference. The reviewer is convinced.
- A question raised about how important to train the VLLM on the human feedback dataset and if it is possible to rely on the generalization capabilities of the base VLLM. Does the refinement model and iterative refinement framework still work if the VTON-VLLM is replaced with an off-the-shelf VLLM and the refinement model is retrained on top of the output of the off-the-shelf VLLM? The authors provided empirical comparisons to address the concern.